# Environment and Local Substrate Availability Effects on Harem Formation in a Polygynous Bark Beetle

**DOI:** 10.3390/insects12020098

**Published:** 2021-01-24

**Authors:** Melissa J. Griffin, Matthew R. E. Symonds

**Affiliations:** Centre for Integrative Ecology, School of Life and Environment Sciences, Deakin University, Burwood, VIC 3125, Australia; matthew.symonds@deakin.edu.au

**Keywords:** harem polygyny, *Ips grandicollis*, environmental effects, mating behaviour

## Abstract

**Simple Summary:**

Harem polygyny is a mating system where a single male defends a group of females for the purpose of securing multiple mating. While this mating system is well-known in mammals it is uncommon in insect groups. The mating aggregations that occur in insect groups may be driven by environmental conditions or resources available for feeding and breeding. We aimed to determine how the local availability of breeding substrate affects the formation of harems in the five-spined bark beetle, *Ips grandicollis*. Aggregations are formed when a male bores under the bark of felled pine trees and makes a nuptial chamber. The male then releases an aggregation pheromone that attracts females for mating and other males to also exploit the resource. When the population density was higher the number of females associated with each male was greater. The population density was determined by environmental circumstances with higher density in a pine plantation that was being harvested than in a plantation that was still standing. The amount of substrate (logs per replicate pile) available to the bark beetles also influences the number of beetles attracted to a log and size of individual harems. The environment and local substrate availability did not affect how females distribute themselves around the male. Females did not actively avoid positioning themselves further from neighbouring females to avoid competition. Their arrangement within harems was equivalent to random positioning.

**Abstract:**

Many forms of polygyny are observed across different animal groups. In some species, groups of females may remain with a single male for breeding, often referred to as “harem polygyny”. The environment and the amount of habitat available for feeding, mating and oviposition may have an effect on the formation of harems. We aimed to determine how the surrounding environment (a harvested or unharvested pine plantation) and availability of local substrate affect the harems of the bark beetle, *Ips grandicollis* (Coleoptera: Curculionidae: Scolytinae). In a harvested pine plantation with large amounts of available habitat, the population density of these beetles is much higher than in unharvested plantations. We found the number of females per male to be significantly greater in the harvested plantation than the unharvested one. Additionally, the amount of substrate available in the immediate local vicinity (the number of logs in replicate piles) also influences the number of beetles attracted to a log and size of individual harems. We also examined how females were distributing themselves in their galleries around the males’ nuptial chamber, as previous work has demonstrated the potential for competition between neighbouring females and their offspring. Females do not perform clumping, suggesting some avoidance when females make their galleries, but they also do not distribute themselves evenly. Female distribution around the male’s nuptial chamber appears to be random, and not influenced by other females or external conditions.

## 1. Introduction

Polygyny is the consequence of males being able to gain access to multiple females in a population within a single breeding season [1]. Species that maintain associations of a single male with a group of females have been described as harem polygynous [1,2]. Best known from mammalian examples, aggregations often occur when females receive some benefit of remaining together as a group, such as reduced harassment from males (e.g., southern elephant seals, *Mirounga leonine*—Carnivora: Phocidae) [3], or where there are drivers such as the availability of roost sites that lead to female aggregations (e.g., Indian short-nosed fruit bats, *Cynopterus sphinx*—Chiroptera: Pteropodidae) [4]. 

The habitat of a species and the potential of the local environment to influence population density may play large roles in the evolution of polygyny and the type of polygynous mating system used by a population or species [5,6]. Some habitats may have a resource that attracts groups of females (e.g., for nesting or oviposition), and some habitats may make it easier to defend groups of females. For example, thicker bamboo stems attract greater numbers of females to mating aggregations in the hemipteran *Notobitus meleagris* (Hemiptera: Coreidae), resulting in larger harems or alternatively greater multimale, multifemale mating aggregations [7]. Likewise, if there are differences in the quality of territories held by males, polygyny could arise because females will choose to mate with males on superior territories. In giant sand treader crickets *Daihinibaenetes giganteus* (Orthoptera: Raphidophoridae), males compete for access to females by defending burrows. However, burrows readily collapse in the sandy environment, and females have a strong preference for association with males with intact burrows [8]. In such circumstances, polygyny should evolve if it is more beneficial for females to breed with an already mated male on a good territory than with an unmated male on a poor territory [5,9]. 

The habitat that an animal lives in may constrain it to a particular mating system. Among seal species (Carnivora: Pinnipedia: Otariidae/Phocidae), the habitat where they mate influences the level of polygyny seen among species. Otariids are more likely to mate on land, and hence the large aggregations of females on beaches allow the males to monopolise groups of females in a harem polygynous mating system. By contrast, in phocids, which are more likely to mate in water, it is much more difficult for males to defend a large group of females and their mating systems shows much less polygyny [10]. The food resources provided by a habitat could also play a role in attracting large numbers of females to an area where males are found. For example, in the African wild donkey, *Equus africanus* (Perissodactyla: Equidae), females are gregarious and found in larger groups in habitats with increased food availability, which allows them to be defended by a dominant male [11]. It is perhaps this habitat and limited mobility of some insects such as bark beetles (Coleoptera: Curculionidae: Scolytinae), and angel insects (Zoraptera: Zorotypidae) living in the closed microhabitat beneath bark that allows aggregations of females to be defended by males [12]. 

In insects, harem polygyny is a relatively uncommon mating system (see Rowell 1987 and Griffin et al. 2019, for discussion of problems associated with defining insects as harem polygynous) [13,14]. However, this mating system has been best studied in bark beetles from the tribe Ipini (Coleoptera: Curculionidae: Scolytinae). These bark beetles differ from other wood-boring insects in that they spend most of their active adult life beneath the bark of dead wood [15]. The adult stage of many other wood-boring insects is free flying and only in the larval stage do they tunnel into wood. The polygynous males from species in the genus *Ips* bore into the bark where they create a small nuptial chamber and release pheromones to attract females [16,17]. Females enter the males’ chamber and bore individual galleries radiating out from the central chamber, where they lay eggs in individual niches. 

In the five-spined pine engraver, *Ips grandicollis*, individual males can be associated with up to seven females at one time [18]. However, harems with four *I. grandicollis* females have the greatest fecundity, showing that the number of females per male affects oviposition behaviour [18]. As harem size increases beyond this, females may suffer reproductive costs, and indeed in the congeneric *I. pini*, mean reproductive success decreases as the density of males and females increases in logs [19]. This may be either because the females receive less paternal care from the males in the form of frass removal [20] or because their larvae encounter greater competition. In such circumstances, females will need to build egg galleries closer together which means that larvae have to compete for food and space, leading to decreased survival [17,21]. Closely spaced egg galleries mean that not all eggs develop, and the competition for phloem available to females and their offspring would suggest that females should try to avoid building their tunnels too close to other females. 

Population density may contribute to the spacing of the female gallery arms in some species. If males are forced to build nuptial chambers close together in high density populations, females may need to make decisions about how to distribute themselves among males [22]. Female *I. acuminatus*, however, do not build their galleries evenly around a male, suggesting that they are not actively avoiding competition with other females [22]. Likewise, Schlyter and Zhang (1996) found no correlation between the distance between female galleries and the size of a harem for each of *Ips typographus*, *I. cembrae* and *I. duplicatus*. These studies, however, only estimated the average distance between females and do not provide information on the specific arrangement of females around a male within each harem size or details about whether females are clumping their galleries in one direction (for example, along the grain of the wood). 

The first aim of this study was to determine the effect of local substrate availability (the amount freshly cut pine logs presented in a pile) on the number of females attracted to individual male *Ips grandicollis*. It would be expected that when these bark beetles have access to more substrate the number of females attracted to each male would decrease as there would be more room available for each male in the immediate vicinity. However, this relationship with local substrate availability may be also affected by the density of beetles in the surrounding environment as a whole, which is in part determined by the amount of suitable habitat that is available in the area. Available habitat increases when pine plantations are logged. *I. grandicollis* typically attacks recently felled trees, therefore when the trees are cut for harvest and a large amount of fresh debris is left on the ground there is potential for a large increase in population size (and hence density). When bark beetles have access to more resources in the environment (e.g., in a harvested plantation), we would expect the number of females that are attracted to each male would increase as there are more individuals in the population. We therefore conducted experiments in both harvested and unharvested pine blocks, experimentally manipulating the availability of substrate (number of logs) in each of these environments. 

Our second aim was to examine whether local substrate availability and the surrounding environment influenced the number of beetles in total that are attracted to logs, and the density of males (hence harems) within the logs. When more substrate is available, it would be expected that more individuals would be attracted to a resource. Initially, more males would be attracted to the log piles (because they are larger) and then more females would be attracted by the increase in the number of males. We also consider whether or not individual females are spacing themselves evenly around a male by examining the distance and angle between female galleries.

## 2. Materials and Methods

*Ips grandicollis* (Coleoptera: Curculionidae: Scolytinae), the five-spined pine engraver, was introduced to Australia from North America in the 1940s and is now a pest of pine plantations throughout the mainland [23,24]. Field manipulations were conducted in *Pinus* spp. plantations at Beerwah, Queensland, Australia (−26.863, 153.040 − 26°51′). The mean daily maximum temperature for September 2017 when this sampling was conducted was 28.9 °C. Billets sampled were harvested from an unmanaged block of pine trees that did not show signs of any previous insect damage. The lengths of the billets ranged from 36 to 43 cm and the diameters (across one end) ranged from 12 to 19 cm. Manipulated piles of billets were set up in one of two environments—the first was an unharvested pine block (46 ha) and the second had been recently felled (42 ha harvested block). The harvested block (between 25 and 30 years old) had been felled within three months prior to the setup of the trial and had a large amount of slash and other pine debris lying on the ground that would have provided additional habitat for bark beetles over the preceding months. The unharvested block was 23 years old and had not been thinned for several years, with very little extraneous dead pine material under the trees, and hence very little suitable habitat for bark beetles in the preceding months. The two blocks were more than 1km apart, considerably greater than the usual dispersal distance for *Ips grandicollis* (median dispersal distance is 0.13 km) [25]. In addition to differences in the surrounding environment, two levels of fresh “local substrate availability” were tested in the manipulated log piles that were set up—low availability (five billets) or high availability (ten billets). These numbers were chosen to balance a clear difference in availability (high availability being double the amount of substrate), while being practicable to replicate and monitor within the time period of initial log colonisation. Five replicate piles of both the low and high resource availability were placed in each of the environment types, making 10 piles in each environment (20 piles in total). The piles were systematically placed at 100 m intervals along a transect just off a logging track, alternating between high (ten billet) and low (five billet) local substrate availability piles to reduce spatial confounding effects. Billets in each pile were stacked on top of each other in pairs alternating the direction at each level, so that beetles would have access to all sides of most of the logs, and each pile had an Ipsenol pheromone lure (IP035-40 mg Ipsenol lure from WestGreen Global Technologies) attached to ensure beetle attraction (Figure 1). Ipsenol is the key constituent of the aggregation pheromone produced by male *Ips grandicollis* to attract conspecifics to overcome tree defences and females specifically to their nuptial chamber for mating.

Logs were left in the field for two weeks. After this time, the bark was removed entirely from each log, being careful to ensure all the galleries were left intact and each male nuptial chamber was numbered. To assess harem size and how this changes with resource availability and the environment/habitat differed, the number of females associated with each male was recorded. The distance from a male’s nuptial chamber to the next closest nuptial chamber was measured, as an indicator of how male spacing adapts to changes in the population density or resource availability. Females also have the opportunity to adapt to different population densities and resource availabilities by changing the distance at which make their galleries, which we also measured. Measurements were made between neighbouring galleries at a standardised distance from the male’s nuptial chamber of 2.5 cm (Figure 2). At this distance, the orientation of the galleries within the phloem is generally set and the proximity to other galleries is consistent. Measurements were made sequentially moving from one female to its neighbour in a clockwise direction around a male’s nuptial chamber. The measured distance, along with the standardised 2.5 cm distance gave an isosceles triangle (see Figure 2) from which we were able to calculate the angle, θ, that the females diverged from each other relative to the nuptial chamber. This is generated by the number of angles for each harem equal to the harem size minus one.

### 2.1. Spacing of Females

We calculated a “proximity index” to measure the extent to which females are clumped in the spacing of their galleries around the male nuptial chamber. The proximity index (PI) for a harem is calculated using the following formula: $PI=\;\frac{\theta max-\theta min}{360-\left(10\;h\right)}\;.\;\frac{dmax-dmin}{4.6}$. Where *h* is the harem size, θmin and θmax are the observed minimum and maximum angles between the galleries in the harem, and dmin and dmax are the observed minimum and maximum distances between individual female galleries. The minimum possible angle between any two females is 10°. This was chosen because in the natural population of *I. grandicollis* studied, no two galleries were within 10° of each other. The value of 4.6 refers to the maximum distance two females could be apart (5 cm) minus the minimum distance—0.4 cm or 10°. The proximity index ranges from 0 to 1, with a proximity index closer to 1 indicating females are highly clumped, and an index of 0 indicating the females are evenly spread around the male. The proximity index was not able to be calculated for harems with only two females as the second angle is often greater than 180°, so the maximum and minimum distances apart will be the same; however, these harems were included when the minimum distance between two females in a harem was analysed (see below). 

### 2.2. Statistical Analysis

We used a generalised linear mixed model with Poisson distribution and log-link function to model the number of females per male and the total number of beetles attracted to each log. For the model predicting number of females in each harem, the predictor variables (factors) were resource type (10 or 5 logs), environment type (harvested versus unharvested plantation), with total surface area of the billet and the mean distance between individual harems on each log as covariates. The random effect was log ID. The variables predicting the total number of beetles per log were resource type, the environment type, and the total surface area of the log. Here, the random effect in the model was tree ID (the tree from which the log was taken). We also used a generalised linear mixed model with Gaussian distribution and identity-link function to model the density of males in a log (males per m^2^) and the distance to the nearest harem (measured between male nuptial chambers). The variables predicting the male density were resource type and environment, and the random effect here was replicate (i.e., each log billet pile). The variables predicting the distance between harems were resource type, environment type and the total surface area of the logs (m^2^). The random effect here was log ID. All GLMMs were run in R [26] using the function glmer in the package lme4 [27]. We also tested candidate models including interactions between the fixed factors. Akaike’s information criterion with correction (AICc) was used to select the best model in each case [28]. For each GLMM, we compared models with all possible combinations of predictors. When more than one model presented an ΔAICc lower than two, we selected the best model as the one with the lowest number of parameters and considered the additional parameters included in the more complex models to be uninformative [28,29]. A normal Q-Q plot was used to determine if the quantiles were evenly distributed in each model. All quantiles show the same distribution.

A permutation test was conducted to compare the observed mean proximity index against expected null distributions of the proximity index. To generate the null proximity index for each harem size (3 to 7), we carried out 10,000 simulations of females being placed randomly in harems (with minimum spacing of 10° between females). The actual mean proximity index for each harem size was compared to the null distribution to calculate a *p* value for the data in relation to the null hypothesis of females distributing themselves randomly (the proportion of values of PI that were equal or more extreme than the observed PI). The minimum distance between two females in a harem was also analysed for all harem sizes (2 to 7 females) using a similar method with a null distribution with 10,000 iterations generated for the expected minimum distance in each harem size if females were spacing themselves randomly. All of these calculations were performed separately for the harems in each of the treatments (5 versus 10 logs, harvested versus unharvested) to assess whether there were different patterns of female spacing in relation to treatment. 

## 3. Results

In total, we collected information from 2461 harems. Harem size varied from one to seven females per male (mean = 3.5 ± 0.02, median = 4). The surrounding environment (harvested or unharvested plantation) had the greatest impact on the number of females associated with each male. The best model predicting harem size (number of females per male) only had a single predictor: whether or not the surrounding pine plantation was harvested. The mean harem size in the harvested environment was 12.4% larger (0.41 females) bigger than the harems in the unharvested environment (Poisson model estimate ± s.e. = 0.1171 ± 0.0217, z value = 5.397, *p* < 0.001; Figure 3A, Appendix A). 

The best model (lowest ΔAICc and simplest model) predicting total number of beetles per log consisted of size (surface area) of log and an interaction between the environment and local substrate availability (Table 1 and Appendix A). Larger logs had more beetles. A pairwise comparison of the best model found that logs in the low local substrate availability (five logs) treatment and unharvested blocks had significantly more beetles per log than logs in low local substrate availability harvested blocks (Figure 3C). However, when local substrate availability was high (10 logs) there was no significant difference in the total number of beetles on each log between harvested and unharvested blocks (z ratio = −1.351, *p* = 0.1768; Table 1).

The amount of local substrate available was the best predictor for the distance between harems. The distance between male nuptial chambers was greater in the piles of 10 logs than the piles of five logs (Figure 3B, Appendix A). The average distance between harems in the five log piles was 9 mm (±1) less than the average distance between harems in the 10 log piles (F-value = 41.79, *p* < 0.001—Figure 3B). The surrounding environment (unharvested or harvested plantation) had no significant effect on the distance between harems. 

The best model predicting the male density on logs included an interaction between the environment and local substrate availability (Table 2 and Appendix A). A pairwise comparison of the best model found that logs in low local substrate availability (5 logs) treatment and unharvested blocks had a significantly higher male density per log than logs in low local substrate availability piles in harvested blocks (*t*-value= −3.937, *p* < 0.001—Figure 3D). However, there was no significant difference in the male density on each log between harvested and unharvested blocks when local substrate availability was high (10 logs) (*t*-value = 1.555, *p* = 0.122).

### Distribution of Females

The proximity index of aggregations in the field experiments ranged from 0.0013 (three females, 10 logs in unharvested block) to 0.6114 (four females, 10 logs in harvested block). The mean observed proximity index of the female galleries around the central male nuptial chamber was not significantly different from the null distribution for any harem size (Table 3). In all cases, the observed mean proximity index was closer to zero, suggesting a tendency to spread out around a male, but not significantly so. This result also held when comparing the harvested and unharvested environments and the five log piles versus the 10 log piles (Appendix A). The observed minimum distance between two females within each harem size was also not significantly different from the mean null distribution of minimum distances between females (Table 4). Again, this result held when comparing the unharvested and the harvested blocks and the harems on the five logs against 10 logs (Appendix A). 

## 4. Discussion

Our study has identified that both surrounding environment and local substrate availability influence the attraction of bark beetles to log, and the allocation of females to harems, but in different ways. The surrounding environment influenced the number of female *Ips grandicollis* associated with each male. Harems were larger in the harvested block regardless of the resource levels (i.e., whether the log piles consisted of 5 or 10 logs). However, the total number of beetles per log and the density of males within a log showed a more complicated pattern in relation to resource type and the surrounding environment, with both being higher when the resource was less abundant (five logs per pile) and the unharvested environment, but lower when the resource was more abundant in the same environment. Surrounding environment therefore plays a key role in determining the effect of local substrate availability on infestation of logs, with the effect being stronger when there are less available resources in the surrounding habitat. Another measure of extent of infestation—the distance between harems (i.e., the distance between male nuptial chambers)—was influenced only by the local substrate availability, with less distance between harems in smaller log piles. 

Despite the overall likely higher population numbers in the harvested plantation, the large amount of additional resources already available in this surrounding environment is likely contributing to the lower density of males and overall number of individuals in the experimental logs. In this environment, there would be a lot of habitat available for the beetles to attack so the additional logs would not be any more attractive to them than the other fallen trees. By contrast, in the unharvested environment with fewer alternative resources available beetles would be much more attracted to new available resources added to their environment. Under such conditions it may be that *I. grandicollis* has to make use of any available resource even if this means that infestation density is higher and some individuals may end up very close together. 

In our experiment, higher insect densities were associated with lower local substrate availability in unharvested environments—individuals are attracted to any suitable habitat. However, by contrast, larger harems were found in the harvested environment. This suggests that the determinants of how beetles join aggregations differ between the sexes in *I. grandicollis*. In females, joining aggregations is more heavily determined by the environment (unharvested or unharvested) while in males, at least in unharvested populations, the major determinant is the local substrate availability (high or low). This difference may be explained by the overall population density. Here, we assume that the overall population density in the environment is high because of the large amount of habitat available in the harvested trees. Consequently, with fewer males per log, females may be more likely to join any harem as they encounter them, rather than risk searching for other males. At high population density, *Dentroctonus ponderosae* bark beetles (Coleoptera: Curculionidae: Scolytidae) were less selective when choosing a tree than at other densities [30]. In another insect often described as “harem polygynous”, the Auckland tree wētā *Hemideina thoracica* (Orthoptera: Anostostomatidae), manipulation of local density significantly affected the composition of male–female aggregations with more females associated with males at higher densities (although not specifically in harems) [31]. On a broader scale, one may expect such an association presumably because males are likely to have more opportunity to encounter additional females at higher population densities [32]. However, such patterns are not inevitable: the population density of chuckwalla lizards had no effect on the number of females associated with each male’s territory [33]. 

Male *I. grandicollis* may be equally attracted to all resources as potential reproductive sites and in the unharvested plantation the local substrate was likely the only new habitat available. Under such circumstances, if a male reaches a log that already has a high density of males it may be too costly to move on or find another habitat. Males are not able to assess the availability of other habitat and risk predation while searching for a new site [34]. By contrast, Zhang et al. (1992) suggested that female bark beetles are able to determine the colonisation density of a log by the concentration of pheromone emitted, before they land. If there is a high density of males on a log, as in the unharvested environment, then females may look elsewhere rather than join the first male she encounters. Habitat cues and the potential for competition with other individuals were most important when the population density was high in *D. ponderosae* [30].

There are other considerations that need to be taken into account when males and females are making reproductive site decisions. At high densities of *I. typographus* and *I. cembrae*, individual females had reduced numbers of offspring [35,36]. Decreased offspring production is consequence of lower oviposition rates and reduced survival of offspring due to high larval competition at high densities [37]. When coupled with reduced substrate availability, this results in eventual overall declines in population density [38]. Bark beetle females could simply be running out of space for their galleries when high numbers of individuals are present, making them only able to build short galleries and lay fewer eggs [36]. This may be partly overcome by females re-emerging earlier from the wood, as at these high densities there was deterioration of the breeding substrate beneath the bark [39]. However, females should theoretically avoid males that are found on densely populated logs. 

Within harems, female *I. grandicollis* tended to arrange themselves randomly around the males’ central nuptial chambers at all harem sizes. We did not find evidence that females were arranging themselves in the most evenly dispersed manner, minimising competition between offspring. They were also not overly clumped, or radiating from one side of the chamber, which might suggest advantages to galleries in particular orientations within the log, such as along the grain of the wood. These results also suggest that there are not large negative consequences of building galleries too close to another female. This relationship also held for all conditions tested in this experiment. Females were arranged randomly in the harvested and the unharvested plantation blocks and when found in piles of 5 versus 10 logs. This result suggests that female spacing behaviour within harems is consistent, irrespective of the harem size, the density of the beetles, and harems within the logs or the surrounding environment.

These results correspond with those found in *I. acuminatus* and *I. typographus*, where females also do not distribute themselves in a manner that would be expected to minimise competition [17,22]. Females may take this potential competition into account when they are laying their eggs. *Ips typographus* laid more eggs on the opposite side of their gallery to where the potential competition is with a neighbouring female’s gallery; additionally, females could build longer galleries to space their eggs out more [17]. Similarly, in other insects that nest in aggregations such as the bumblebee wolf *Philanthus bicinctus* (Hymenoptera: Crabronidae) where males monopolise and defend females [40], the main considerations for nest sites of females is the proximity to their site of emergence regardless of how close to other females they are [40,41]. By contrast, vertebrate female aggregations do display active spacing of individuals. In cichlid fish *Lamprologus ocellatus* (Cichliformes: Cichlidae), females that settle first act aggressively towards incoming females attempting to exclude them from a male’s territory. New females compete for male parental investment and the offspring of one female are potential predators of another’s offspring [42]. If active competition was happening in the *I. grandicollis* population, we would expect to see females spacing themselves more evenly around males or females may be deal with this through spacing of eggs. 

It is unlikely that females are able to detect other females that are boring nearby either within their harem or in neighbouring harems [17,22]. Females may instead use other cues to make decisions about the male nuptial chamber they enter and where they will build their galleries. Females can use different cues as they enter a male’s nuptial chamber depending on the number of females already associated with this male [43]. Pheromones are important in bark beetle populations. Males use pheromones to attract females to the wood where they have made their nuptial chambers; there is potential for females to use these pheromones to provide information on male quality. Harem size may also be affected by the quality of the wood and phloem layer. *Ips calligraphus* has larger harems in thick phloem layers [44]. Predators have been shown to suppress population numbers in bark beetle species and play a role in their population dynamics [45,46,47]. Hence, females may trade off costs of dispersal with staying at the first log they land on and the increased costs of breeding at high densities [19]. Previous studies of *I. grandicollis* have indicated that females choose males at random and do not select on the basis of quality of the male (e.g., male size) or the number of females already associated with the male [18,48,49]. Random selection of bore holes may be a consequence of predator avoidance by females by selecting the first available hole that they encounter [17]. 

## 5. Conclusions

This study provides a snapshot of the mating behaviour in *I. grandicollis* comparing two Australian pine plantations in the same area. External factors such as climate, time of season, and harvesting history of surrounding blocks will likely have an effect on the formation of mating aggregations that we have not been able to consider here. However, despite these confounds, this study does show clear patterns in the effect of environment and local substrate availability. Male and female *I. grandicollis* may use different cues to choose their habitat and where they are going to breed. In the assumed high population density in the harvested plantations, more females are attracted to each male. Males are at the highest density within logs when there is only a limited amount of available habitat irrespective of the overall population density. However, once the females were inside a male’s nuptial chamber, the females neither clump nor evenly space themselves. It is possible that females use other methods, such as placement of eggs [36], to ensure the greatest number of larvae survive. This may be further evidence that mating aggregations are formed with competing interests of males and females. Finally, it is notable that large densities of beetles were still seen to occupy logs found in the unharvested environment. This indicates that a single windfall event in an unharvested population could provide enough substrate to attract a high density of beetles. These results suggest that unharvested plantations do not necessarily have reduced risk of beetle attacks and should still be monitored with good silvicultural practices by forestry companies appropriately.

## Figures and Tables

**Figure 1 insects-12-00098-f001:**
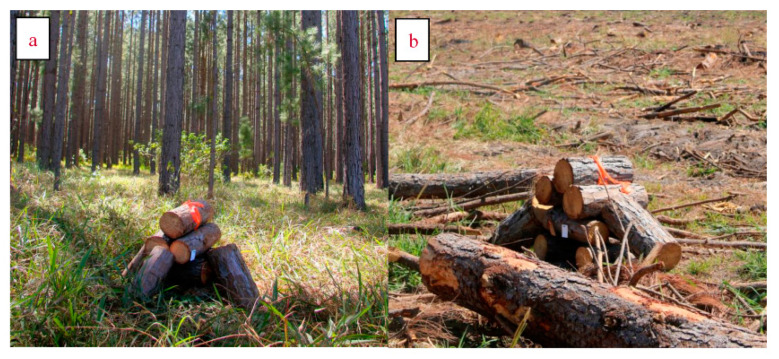
Set up of the local substrate availability in the unharvested (**a**) and harvested (**b**) pine plantation.

**Figure 2 insects-12-00098-f002:**
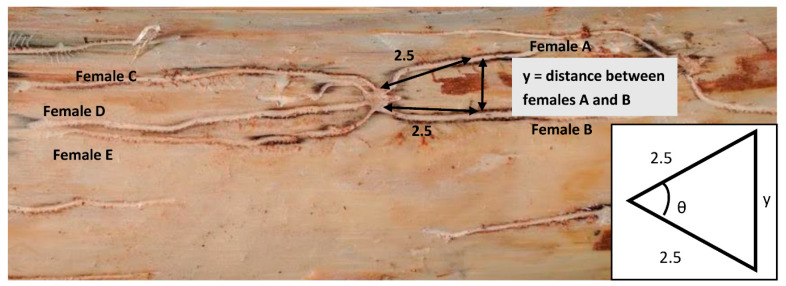
Example of *Ips grandicollis* harem showing a central male nuptial chamber and the galleries of five females radiating outwards. The arrow marked with a y indicates where measurements between two females were taken from. θ = the angle calculated in to incorporate into a proximity index (see below).

**Figure 3 insects-12-00098-f003:**
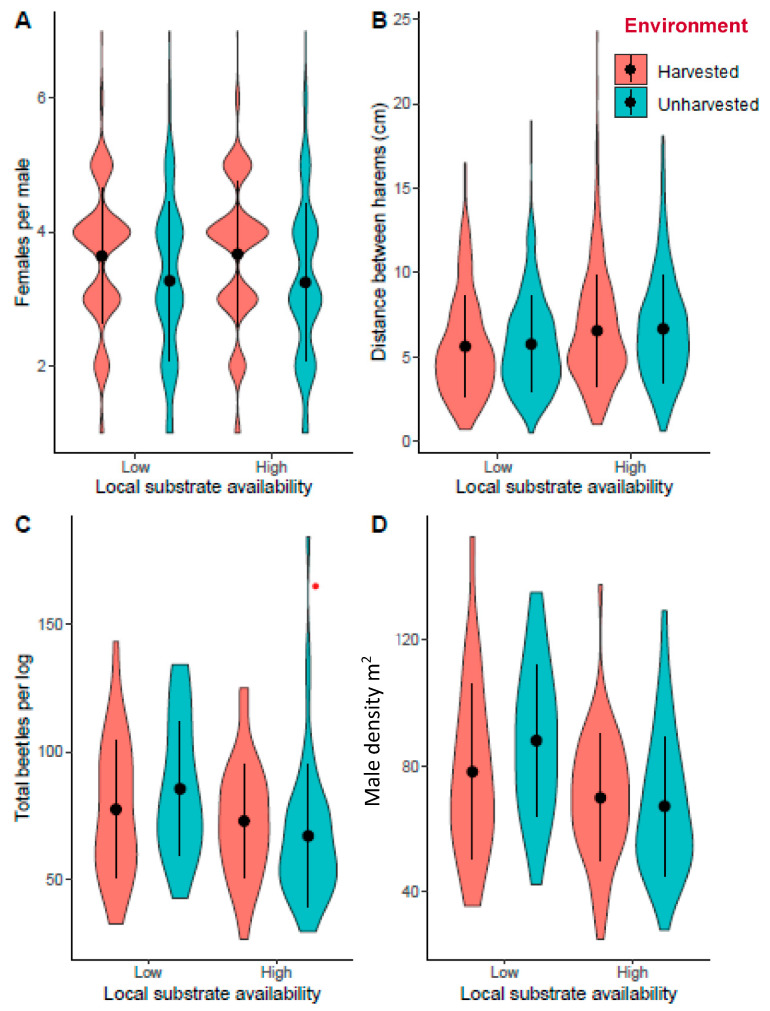
Violin plots indicating the spread and (**A**) mean (±SD) number of females per male (harem size), (**B**) mean (±SD) distance to the nearest harem (male nuptial chamber), (**C**) mean total number of beetles (males and females) per log (±SD), and (**D**) mean density (±SD) of males on a log (number of males per m^2^) in relation to local substrate availability (high = 10 logs/pile, low = 5 logs/pile) in harvested and unharvested pine plantations.

**Table 1 insects-12-00098-t001:** Best generalised linear mixed model selected to predict the total number of beetles found in a log. Model estimates for categorical predictors are those for the condition represented in parentheses.

Predictors	Estimate	SE	*Z*-Value	*p* Value
Local substrate availability (5 logs)	0.10544	0.02838	3.715	<0.001
Environment (unharvested)	−0.06491	0.02432	−2.669	0.008
Total surface area	3.16424	0.42990	7.360	<0.001
Local substrate availability (5 logs) × Environment (unharvested)	0.10757	0.03986	2.699	0.007

**Table 2 insects-12-00098-t002:** Best generalised linear mixed model that predicts the density of males (males per m^2^) in a log.

Predictors	Estimate	SE	F-Value	*p* Value
Environment (unharvested)	−2.709	6.474	0.3505	0.5689
Local substrate availability (5 logs)	8.257	5.309	15.08	<0.001
Local substrate availability (5 logs) × environment (unharvested)	12.642	7.508	2.835	0.0945

**Table 3 insects-12-00098-t003:** Summary table of the observed and null mean proximity index of females within a harem, for each harem size of 3 to 7 females per male across all treatments. Proximity index closer to 1 indicates females are highly clumped. While an index of 0 indicates the females are evenly spread around the male.

Harem Size	*N*	Observed Mean	Null Mean	*p* Value
3 females	726	0.189	0.283	0.3941
4 females	831	0.217	0.301	0.3260
5 females	343	0.211	0.310	0.2311
6 females	58	0.183	0.301	0.1473
7 females	9	0.191	0.288	0.1629

**Table 4 insects-12-00098-t004:** Summary table of the observed and null mean minimum distance (cm) between two females within a harem, for each harem size of 2 to 7 females per male across all treatments.

Harem Size	*n*	Observed Mean	Null Mean	*p* Value
2 females	384	3.47	2.20	0.5663
3 females	726	2.27	1.98	0.5380
4 females	831	1.74	1.72	0.5722
5 females	343	1.29	1.56	0.6237
6 females	58	1.03	1.47	0.5780
7 females	9	0.72	1.38	0.6216

## Data Availability

The data presented in this study are available on request from the corresponding author. Output from all generalised linear models and the breakdown of the proximity index are available within the article and in the Appendix A.

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
