# Peer review of "Environment and Local Substrate Availability Effects on Harem Formation in a Polygynous Bark Beetle"

_insects, 2021, doi:10.3390/insects12020098_

Round 1

Reviewer 1 Report

This is an interesting paper that reports the effects of habitat type and resource abundance on Ips grandicollis harem formation and gallery construction patterns. I have noted a few points that require correction in the main text, clarification, or justification.

There are typos / grammatical mistakes in the Simple Summary (lines 11, 14, 19).

Materials and Methods

It would be helpful to have an illustration of the experimental set up. Lines 139-141 are slightly confusing. An image would help here. What were the sizes of the unharvested and harvested pine blocks?

Why use 5 or 10 logs for “low” or “high” resource availability, respectively? Was the choice arbitrary or was it based on sound criteria? How does it relate to natural conditions? The authors need to justify their selection.

What was the distance between the ‘replicates’ in the 5 or 10-log set-up? Similarly, what was the distance between 5-log piles and 10-log piles within a given block? How will the interaction between log piles affect the preferences of I. grandicollis? In fact, it appears that the ‘replicates’ are actually not independent (pseudoreplication) which violates the assumption of independence between data points invalidating/introducing errors in the statistical analysis. There aren’t sufficient independent blocks for generalization of the results.

Field sampling was conducted over a 14-d period. Potential influences of inter-annual variation and habitat heterogeneity among multiple sites over a larger scale (only one site each for harvested vs. unharvested type was used in this study) were not considered. How will this limited spatio-temporal scale affect the broader conclusion (i.e., at the population level) regarding the colonization behavior of I. grandicollis?

Could temperature influence harem formation and gallery construction patterns? Could the patterns be different, say if the temperatures are cooler than normal?

Line 142: Ipensol?

Line 169: Equation and parameters need to be corrected.

Results and Discussion

Line 232: In the Predictors column, what does the category of resource availability or environment (in parenthesis) represent? If it is the reference level, please mention this.

In general, I’d suggest that comparing and contrasting mating systems should be restricted to insects. Please minimize or remove the comparison of I. grandicollis harem formation behavior with vertebrates (same with the Introduction section).

Reviewer 2 Report

Review of Insects 1005104 “Habitat and resource availability effects on harem formation in a polygynous bark beetle”

This paper studied how habitat and resource availability affected the formation and distribution of harems in bark beetles. The authors found that there were larger harems in higher quality habitats (harvested plantations), and that females appear to distribute themselves randomly around the male.

Major comments:

My main concern with the paper is that the environmental treatment or habitat (i.e., harvested or unharvested plantations) directly affects the amount of available resources. The authors even say so themselves: “Available habitat increases when pine plantations are logged” (line 106) and “When bark beetles have access to more resources in the environment (e.g., harvest plantation)…” (lines 109-110). I therefore found the term “resource availability” (which the authors used to describe the experimental logs that were set out) very confusing and misleading. More importantly, it is unclear if the experimental logs are a biologically relevant manipulation of “resource availability”, especially given the much greater disparity in resources between the harvested and unharvested habitats. Indeed, the fact that the only significant predictor of harem size was habitat suggests that this variable swamps the effect of “fresh” resource availability.

Along the same lines, I have lots of questions about the design of the study plots. How far apart were the harvested and unharvested plantations? Were they far enough apart that the beetles couldn’t fly between the two? Within each habitat, how far apart were the logs spaced? Would it be feasible for beetles to move from one log to the next (e.g., if they found one log too crowded), or were they spaced too far apart? Why did you choose treatments of 5 log piles and 10 piles? Are these within the range of “fresh resource availability” that the beetles might experience in the wild? Would a beetle in an unharvested plantation ever be expected to experience such abundant resources as 10 logs? If not, can we be confident that their behaviors are normal? I also don’t know what you mean by “stacked on top of each other in pairs alternating direction”. Are they stacked lengthwise? Why do you need to alternate direction? Why in pairs?

How much Ipsenol pheromone lure was applied? Was the same amount applied to each log? Or was the amount adjusted to the size of the log? If it is an aggregation pheromone produced by males, does it deter other males from establishing in the log? I am concerned that using this pheromone could have a big impact on male behavior.

Lastly, I appreciate that sometimes it is beneficial to report the behaviors of an organism, even if there isn’t an immediately obvious reason why the information is important. However, I think it would be helpful to do a better job explaining to the reader how/why this study is important. Can it help us do a better job of managing forests and preventing bark beetle attacks?

Specific comments:

Methods: Why was 2.5 cm used as a standard length?

In figure 1, it would be really helpful to add angles to the diagram.

Similarly, it would be really helpful to include figures of different harems with a low and high proximity index so readers can have a clearer sense of what the difference looks like. What was the lowest and highest proximity index observed?

Lines 358-359: Why can’t females detect other females? I would think that they might be able to detect their odors or maybe also vibrations?

Reviewer 3 Report

The manuscript describes an experiment designed to investigate how various factors may influence the harem size and female egg gallery orientation in the bark beetle Ips grandicollis. Studies on how harems are formed and how costs and benefits influence the number of accepted or accepting females have been performed on a variety of animals, some of which are exemplified by the authors. One question that always can be raised when polygynous mating systems are discussed is “where do all the males go?” So were there a lot of male-only nuptial chambers found in the logs, or can the missing males be assumed to have been lost during dispersal. When bark beetles attack standing trees, many die due tree defence mechanisms, like resin flow. But, as I understand I. grandicollis only rarely go for those trees. The study of harem size and its dynamics in bark beetles is challenging, as the process is hidden until destroyed, when only a snapshot is revealed. The study presents a large number of such snapshots shoing the variation in female numbers and in their within-harem spacing. This might be a valuable background for future detailed investigations about this or other polygynous bark beetles.

Indeed, there seems to be a difference for some of the variables both between resource availability classes and habitats, although they are mostly small. Therefore, it is difficult to get a clear picture of the results and what the take home message is. I think one reason may be that the expected influence of habitat is not so clear. The cut area obviously offered a lot of suitable breeding material directly after the tree felling, and it is understood that this habitat had a higher beetle density also when the experiment was performed three months later. I suppose the offspring of the beetles infesting the wood after the cutting had emerged by then. It is not clear to me what might be most important for the outcome of the experiment in the cut area: the surplus of breeding material making the experimental bolts comparatively less attractive (despite some of the material had been used during the three months), or the higher population density resulting in higher attack densities. Maybe these effects even out each other? It is also not clear to me what to expect when it comes to number of females per male and the between-male variation in relation to attack density. On row 103, you argue for decreased number of females per male when density is low, but I would promote the opposite. When male density is low, there is more space to accept more females. Related to this: Is it known how much influence each sex has regarding female entering a harem? The second issue that might have made the results less clear is the small difference in terms of resource availability, only a factor two. A larger difference between low and high would also most likely give more distinct results. A third complicating factor is that the synthetic pheromone seems to have been in place during the entire period. This affected the total pheromone dose released from each resource unit and may have influenced number as well as sex ratio of attracted, landing and attacking beetles. The vague results, probably due to an unlucky experimental design, make the discussion very speculative with some hand-waving, and has to be rewritten.

The manuscript reads well, with the exception of “Simple Summary”, which seems to have been put together in rush. Please rewrite! In the last sentence of this and of the Abstract, the expression “wider environment” is used, but it is not explained what it means. Additional comments by row number:

73: “of their active adult”, since many Ipini spend most of their adult life hibernating.

80-81: Is it possible to talk about the fecundity of a harem?

154: Figure 1

Figure 1: Why all the ¶? Ugly!

169: The thetas in the equation are shown as ?? in the pdf I downloaded. The description of PI is indeed tricky to follow. I would appreciate if a more pedagogic presentation is possible to provide. Is this a new invention or are references available?

194: Please write out what/who [24] is.

Results: In the text figure 2B and 2C have been mixed up. Some Tables are in the main text an some in supplementary information, what determines their position?

Figure 2: Why is the x-axis reversed, from high to low? In the Environment legend, the colours have a pattern, but not in the actual graphs. Please indicate the statistics in some way. In the y-axis of 2D, there is: m

275: This sentence does not make sence. There has to be some specific characteritics in the environment that has positive effect.

297-300: A bit humanized about decisions.

433, 502: Are 2019 a and b necessary?

447, 449: Double numbering.

464: pages missing.

467: Year in wrong place.

489: angew.

Round 2

Reviewer 1 Report

Order and family of certain insect species are missing.

Lines 100, 101, 103: Pinniped species? Otariids? Phocids?

109: Coleoptera: Curculionidae: Scolytinae

Only the first letter of Genus name after the first mention.

Line 183: Perhaps indicate the coordinates in degrees, minutes, seconds or decimal degrees with °E (longitude) and °S (latitude).

Please indicate the dispersal distances that Ips grandicollis can travel. According to Costa et al., 50% of beetles travel beyond 0.13 km.

Line 210: remove ‘to’

Billets were placed at 100 m intervals per plantation. But this is within the median dispersal distance of I. grandicollis of 0.13 km (Costa et al. 2013)?

Reference

Costa A, Min A, Boone CK, Kendrick AP, Murphy RJ, Sharpse WC, Raffa KF, Reeve JD. Dispersal and edge behaviour of bark beetles and predators inhabiting pine plantations. Agric. Forest Entomol. 2013. 15, 1-11.

Reviewer 2 Report

I think the authors have done a sufficient job of addressing my previous concerns.